# Significance of Necroptosis in Cartilage Degeneration

**DOI:** 10.3390/biom14091192

**Published:** 2024-09-21

**Authors:** Md Abdul Khaleque, Jea-Hoon Kim, Md Amit Hasan Tanvir, Jong-Beom Park, Young-Yul Kim

**Affiliations:** 1Department of Orthopedic Surgery, Daejeon St. Mary’s Hospital, The Catholic University of Korea, Seoul 06591, Republic of Korea; abdulkhaleque.dream@gmail.com (M.A.K.); superbdoc@hanmail.net (J.-H.K.); tanvir002@catholic.ac.kr (M.A.H.T.); 2Department of Orthopedic Surgery, Uijeongbu Saint Mary’s Hospital, The Catholic University of Korea, Seoul 06591, Republic of Korea; spinepjb@catholic.ac.kr

**Keywords:** cartilage degeneration, necroptosis, OA, RA, trauma

## Abstract

Cartilage, a critical tissue for joint function, often degenerates due to osteoarthritis (OA), rheumatoid arthritis (RA), and trauma. Recent research underscores necroptosis, a regulated form of necrosis, as a key player in cartilage degradation. Unlike apoptosis, necroptosis triggers robust inflammatory responses, exacerbating tissue damage. Key mediators such as receptor-interacting serine/threonine-protein kinase-1 (RIPK1), receptor-interacting serine/threonine-protein kinase-3(RIPK3), and mixed lineage kinase domain-like (MLKL) are pivotal in this process. Studies reveal necroptosis contributes significantly to OA and RA pathophysiology, where elevated RIPK3 and associated proteins drive cartilage degradation. Targeting necroptotic pathways shows promise; inhibitors like Necrostatin-1 (Nec-1), GSK’872, and Necrosulfonamide (NSA) reduce necroptotic cell death, offering potential therapeutic avenues. Additionally, autophagy’s role in mitigating necroptosis-induced damage highlights the need for comprehensive strategies addressing multiple pathways. Despite these insights, further research is essential to fully understand necroptosis’ mechanisms and develop effective treatments. This review synthesizes current knowledge on necroptosis in cartilage degeneration, aiming to inform novel therapeutic approaches for OA, RA, and trauma.

## 1. Introduction

Cartilage, a tough and flexible tissue at the ends of bones, absorbs shock and reduces joint friction, lacking blood vessels and nerves, so it does not feel pain. Chondrocytes, the specialized cells within cartilage, receive nutrients through diffusion within a gel-like matrix [1,2,3,4]. Damage to cartilage and subchondral bone can result from trauma, arthritis, and sports injuries, affecting about 60% of knee arthroscopy patients and around 15% of people over 60 [5,6]. Cartilage degeneration is a key feature of osteoarthritis (OA), the most common joint disease, leading to significant pain, disability, and socioeconomic costs worldwide [7,8,9,10]. The epidemiology of OA is complex, involving genetic, biological, and biomechanical factors. Chondrocytes, the only cell type in articular cartilage (AC), are responsible for the synthesis and maintenance of the extracellular matrix (ECM) and regulate its enzymatic breakdown [11,12,13]. The loss of homeostasis-favoring catabolic activities results in the progressive destruction of AC, characteristic of OA. Chondrocyte death is considered a crucial event in the pathogenesis of OA [14,15,16,17,18,19]. Clinically, OA can be categorized into inflammatory and non-inflammatory forms. The non-inflammatory subtype, often referred to as degenerative arthritis, typically lacks significant inflammatory cell infiltration. Conversely, inflammatory OA involves cellular inflammation mediated by activated leukocytes, which exacerbates cartilage degradation [12,20]. Central to the pathophysiology of OA is an imbalance in anabolic and catabolic processes within the articular cartilage, hampering chondrocyte cells’ ability to synthesize ECM components while promoting enzymatic degradation of matrix molecules, initiating an inflammatory cascade [21,22,23,24]. In OA, a state of low-grade inflammation prevails, characterized by the release of inflammatory cytokines, chemokines, reactive oxygen species (ROS), and matrix metalloproteinases (MMPs) from chondrocytes and surrounding cells [25,26]. Proinflammatory factors such as interleukin (IL)-1β, tumor necrosis factor (TNF)-α, IL-6, IL-15, IL-17, and IL-18, prominently present in cartilage, bone, and synovial tissues, drive the upregulation of MMP expression. Consequently, chondrocytes produce additional inflammatory mediators like nitrotyrosine and IL-6, further amplifying deleterious cellular responses [27,28].

In the early to mid-1990s, there was a common belief that cell death could be classified into two main types: programmed cell death (PCD), which includes apoptosis and autophagy, and an unprogrammed, passive form called necrosis [29,30,31,32,33]. Each type has specific morphological and biochemical characteristics. For a long time, apoptosis and autophagic cell death were considered the only tightly regulated and direct forms of cell death, whereas necrosis was viewed as an uncontrolled process marked by the release of cellular contents, organelle swelling, and plasma membrane rupture [31,34,35,36]. Recent advancements in cell death research have challenged this perspective, revealing that necrosis can also be a highly regulated process known as necroptosis. Necroptosis is primarily controlled by specific genetic factors, such as RIPK1, RIPK3, and MLKL, which have gained attention in cartilage degeneration studies [37,38,39].

Furthermore, necroptosis and apoptosis, although sharing certain signaling pathways, exhibit distinct execution processes in cell death. Both necroptosis and apoptosis can be initiated through common pathways, such as the activation of RIPK1 and the involvement of death receptors [40,41,42]. However, the downstream consequences diverge significantly. In apoptosis, the cell undergoes a controlled and orderly dismantling, leading to cell death without eliciting an inflammatory response [41,43,44]. This process involves cellular shrinkage, chromatin condensation, and the formation of apoptotic bodies readily engulfed by neighboring cells. Conversely, necroptosis culminates in a form of programmed necrosis that triggers inflammation and immune responses [35,41]. Therefore, while necroptosis and apoptosis may utilize overlapping pathways initially, their ultimate outcomes and immunological consequences are distinct [29,41] (Figure 1).

## 2. Relationship between Necroptosis and Cartilage Degeneration

The interrelationship between necroptosis and cartilage degeneration is pivotal in understanding osteoarthritis (OA) pathophysiology. Necroptosis, a regulated form of cell death mediated by receptor-interacting protein kinases (RIPK) 1 and 3 and the mixed lineage kinase domain-like pseudokinase (MLKL), involves cell swelling, membrane rupture, and the release of cellular contents, triggering inflammatory responses. Unlike apoptosis, necroptosis is highly inflammatory, which makes it critical in chronic inflammatory diseases, including OA [45,46,47,48]. In osteoarthritis (OA), chondrocytes, the primary cells in cartilage, undergo necroptosis when exposed to inflammatory cytokines like TNF-α and IL-1β. This process leads to a loss of cartilage integrity and function. Elevated inflammatory mediators in osteoarthritic joints induce necroptosis via TNF receptor 1 (TNFR1), recruiting RIPK1, RIPK3, and MLKL [49,50,51]. Necroptosis degrades the cartilage matrix by releasing DAMPs, which not only exacerbate inflammation and increase MMP activity but also trigger the release of proinflammatory cytokines from chondrocytes, perpetuating joint tissue damage [51,52,53]. Necroptosis is notably more immunogenic than apoptosis due to the robust inflammatory response from DAMPs, leading to heightened extracellular matrix (ECM) degradation [54]. Elevated RIPK3 expression in OA patients underscores the pathway’s relevance in disease progression, with necroptosis’s tissue-destructive potential surpassing apoptosis [55,56].

Despite advancements in understanding OA, current treatments like NSAIDs and opioids offer only symptomatic relief, and surgical interventions become necessary as the disease progresses [57,58]. The multifaceted nature of OA highlights the need for novel therapeutics that halt disease progression and treat early-stage OA [27,41]. Inhibitors targeting RIPK1 and RIPK3 have shown promise in preclinical models by reducing chondrocyte death and preserving cartilage integrity [59,60]. Understanding necroptosis in cartilage degeneration highlights potential therapeutic targets for managing OA. Targeting key components of the necroptotic pathway may offer new strategies to mitigate cartilage damage and ameliorate OA symptoms. This article elucidates the significance of necroptosis in cartilage degeneration, aligning with recent recommendations on the specific usage of “necroptosis” to describe this regulated necrotic process (Figure 2).

## 3. Typical Molecular Mechanisms and Proteins Involved in Regulating Necroptosis

The cellular events acting downstream of the necrotic signaling complex to execute necroptosis depend on the cell type and stimulus. The initiation of the necroptosis signaling pathway is prompted by the activation of death receptors, including the tumor necrosis factor (TNF) receptor, FAS, Toll-like receptor, or interferon receptor [61,62,63,64]. Downstream of these receptors, RIPK1 or other proteins containing RIPK homologous interaction motif (RHIM) domains interact with RIPK3 to form a complex known as the necrosome, leading to the activation of RIPK3 via phosphorylation. Once phosphorylated, RIPK3 subsequently catalyzes the phosphorylation of MLKL, triggering it’s activation [9,56,65,66,67]. Phosphorylated MLKL undergoes oligomerization and translocation to the plasma membrane, culminating in necroptosis. Thus, serine/threonine kinases RIPK1 and RIPK3, along with MLKL, constitute the core machinery of TNF-induced necroptosis [68,69,70,71,72]. In addition to its involvement in the TNF-induced necroptosis pathway, RIPK3 can activate Ca^2+^/calmodulin-dependent protein kinase II (CaMK II), initiating the opening of the mitochondrial permeability transition pore (mPTP), ultimately leading to necroptosis in cardiomyocytes [73,74,75,76,77]. Furthermore, RIPK3 can be activated not only by RIPK1 but also by other proteins possessing RHIM domains, such as TIR domain-containing adaptor protein (TRIF) and DNA-dependent activator of interferon regulatory factors (DAI), thereby broadening the spectrum of activation mechanisms for RIPK3 in necroptosis [41,78,79,80,81] (Figure 3).

## 4. Significance of Necroptosis to Cartilage Degeneration

Necroptosis, a programmed form of necrosis or inflammatory cell death, has emerged as a significant factor in the pathology of cartilage degeneration [41,82]. Understanding the role of necroptosis in cartilage health and disease offers potential insights into novel therapeutic strategies for conditions such as OA. Here are the key aspects highlighting the significance of necroptosis in cartilage degeneration: mechanistic insights, inflammatory response, chondrocyte death, crosstalk with other cell death pathways, therapeutic potential, and disease models and research. To maintain coherence and facilitate discussion, the factors stimulating cartilage degeneration can be categorized into three classes: osteoarthritis (OA), rheumatoid arthritis (RA), and trauma-induced arthritis. Other autoimmune-induced arthritis can be discussed together with RA (Figure 4).

### 4.1. Significance of Necroptosis-Mediated Cartilage Degeneration in Osteoarthritis and Temporomandibular Joint Osteoarthritis

OA and temporomandibular joint osteoarthritis (TMJOA)-induced necroptosis associated with cartilage degeneration have revealed several key mechanisms and therapeutic targets for inhibiting necroptosis-mediated cartilage degeneration, supported by experimental evidence. The study by He, F., et al., identifies a vicious cycle in TMJOA where TNFα induces Syndecan 4 (SDC4), amplifying TNFα signaling and triggering necroptosis in chondrocytes. This necroptosis releases cartilage-degrading enzymes, with SDC4 acting as DAMPs to further intensify inflammation and cartilage damage. Using a model combining Unilateral Arthritis Condition (UAC) and estrogen treatments, the study shows that inhibiting RIPK3, pMLKL, and SDC4 reduces necroptosis, protects cartilage, and reduces inflammation, suggesting that targeting the TNFα-SDC4-RIPK3-pMLKL pathway offers potential therapeutic strategies for managing TMJOA [83,84]. Van der Kraan, Davidson, and van Den Berg identified the role of bone morphogenetic protein 7 (BMP7) in OA progression. Cheng, J. et al. revealed that BMP7 induces chondrocyte necroptosis in a dose-dependent manner. Silencing BMP7 reduces RIPK1-induced necroptosis and restores ECM gene expression, demonstrating that BMP7 overexpression correlates with increased necroptosis and cartilage loss in OA models. This highlights a novel MLKL-independent mechanism where RIPK1 drives necroptosis through BMP7, suggesting that targeting the RIPK1-BMP7 interaction could offer new therapeutic approaches for OA treatment [85,86]. In another study, Jeon, Jimin, et al., discovered that RIPK3 expression is significantly higher in damaged cartilage of OA patients. Their mouse model demonstrated that RIPK3 overexpression accelerates cartilage degradation, while RIPK3 depletion reduces OA severity. Knocking down TRIM24, a negative regulator of RIPK3, increased RIPK3 expression and promoted OA progression [87]. They identified AZ-628 as a potent RIPK3 inhibitor through the Connectivity Map (CMap) approach and in silico binding assays, showing that AZ-628 effectively mitigates RIPK3-mediated OA pathogenesis and suggesting its potential as a therapeutic agent for OA [24,88]. Sun, Kai, et al., found that inhibiting TRADD with ICCB-19 ameliorates chondrocyte necroptosis and OA by blocking the RIPK1-TAK1 pathway and restoring autophagy. ICCB-19 protected chondrocytes from TNF-α-induced damage, reducing inflammation, ECM degradation, and cell death. In vitro, TRADD inhibition decreased TNF-α-induced necroptosis and inflammation, while in vivo ICCB-19 injections halted cartilage degeneration in a DMM-induced OA model, indicating its potential for clinical application [89]. Chen, Xiaolei, et al., discovered that PLCγ1 inhibition combined with blocking apoptosis and necroptosis enhances cartilage matrix synthesis in IL-1β-treated rat chondrocytes. Their rat OA model showed that low doses of the PLCγ1 inhibitor U73122 improved Collagen2 and Aggrecan levels, while high doses increased chondrocyte apoptosis and necroptosis. Combining U73122 with Z-VAD and Nec-1 effectively increased matrix synthesis, suggesting that combination therapy targeting PLCγ1, apoptosis, and necroptosis may be a promising OA treatment strategy [90]. Liang, Shuang, et al., demonstrated that decreased RIPK1 expression alleviates OA by disrupting the TRIF/MyD88-RIPK1-TRAF2 negative feedback loop. Their in vivo experiments showed that RIPK1 phosphorylation is elevated in IL-1β-treated chondrocytes and OA mouse models, and RIPK1 knockdown reduced cartilage damage and inflammation. This modulation of the TRIF/MyD88-RIPK1-TRAF2 pathway highlights RIPK1 as a critical OA regulator and suggests that targeting RIPK1-mediated inflammatory pathways could offer new therapeutic strategies for OA [14,87]. Similarly, Cao, Xin, et al. demonstrated that RIPK1 inhibition in TMJOA models protects cartilage from degradation by suppressing inflammatory signaling and necroptosis, supporting RIPK1 as a therapeutic target [91]. Lastly, Qiu, Jianxin, et al. identified AZD8330 as a promising OA therapy, showing in both in vitro and in vivo studies that AZD8330 activates cIAP1 to inhibit the RIPK1-associated necrosis signaling pathway, effectively preventing chondrocyte necroptosis and preserving cartilage integrity [92]. Collectively, these studies highlight that targeting molecules such as TNF-α, SDC4, RIPK1, RIPK3, BMP7, and PLCγ1 offers promising strategies for reducing necroptosis and inflammation, which are critical for advancing OA and TMJOA treatments (Figure 5) (Table 1).

### 4.2. Significance of Necroptosis-Mediated Cartilage Degeneration in RA

Rheumatoid arthritis (RA) is a chronic autoimmune condition that primarily affects the joints, leading to progressive inflammation. This condition is marked by the infiltration of inflammatory cells into the affected joints, which can result in ongoing damage to cartilage over time [93,94]. Numerous studies have been conducted to identify critical mechanisms and therapeutic targets for managing necroptosis-mediated cartilage degeneration in RA. In this regard, Chen, Yong, et al. demonstrated that necroptosis in RA chondrocytes, driven by high levels of RIPK1, RIPK3, p-MLKL, and PGAM5, can be inhibited by Nec-1 and amiloride. These inhibitors protect cartilage and reduce proinflammatory cytokines by targeting the RIPK1/RIPK3/p-MLKL pathway. Acid-sensing ion channel subunit 1 (ASIC1a) mediates this necroptosis, and its upregulation is reversible by PcTx-1 or Nec-1. ASIC1a, highly expressed in chondrocytes, responds to acidosis from lactic acid accumulation, contributing to RA and joint destruction [95,96,97,98]. Another study by Lee, Seung Hoon, et al. found that interferon-gamma (IFN-γ) can mitigate necroptosis and inflammation in RA by reducing MLKL production and modulating inflammatory responses. IFN-γ decreases necroptosis mediators (RIPK1, RIPK3, MLKL) in collagen-induced arthritis (CIA) mice and RA patients’ synovium. The absence of IFN-γ worsened inflammation through enhanced STAT3 activation, T17 cell differentiation, and elevated TNF-α and IL-17 levels. Despite its proinflammatory role, IFN-γ reduced cartilage damage and downregulated cFLIPL expression, suggesting its therapeutic potential in RA by inhibiting necroptosis and reducing inflammation [99]. Wang, Qiong, et al. demonstrated that KW2449 effectively ameliorates CIA by inhibiting RIPK1-dependent necroptosis. Compared to methotrexate (MTX), KW2449 significantly reduced joint swelling, arthritis scores, and plasma inflammatory cytokines in CIA rats. Histopathology and micro-CT confirmed that both treatments alleviated joint damage and bone destruction. KW2449 uniquely inhibited RIPK1-mediated necroptosis, reducing RIPK1 and MLKL levels and their phosphorylation. Flow cytometry and AO/EB staining revealed that KW2449 primarily reduced necroptotic cells, enhancing cell viability. Knockdown experiments underscored RIPK1’s critical role, highlighting KW2449’s potential as an RA therapeutic by targeting the RIPK1 pathway [100]. Raafat Ibrahim et al. explored irisin’s role in experimentally induced arthritis, focusing on its impact on high-mobility group box 1 (MGB1)/ monocyte chemoattractant protein 1 (MCP1)/Chitotriosidase I–mediated necroptosis. Irisin reduced necroptotic signaling and inflammation by downregulating TNF-α, MCP1, and HMGB1. It promotes chondrocyte recovery and exhibits anti-inflammatory and antioxidant effects through the nuclear factor kappa B (NF-kB) and nuclear factor erythroid 2-related factor (Nrf2) transcription factor/Hemoxygenase 1 (HO-1(Nrf2/HO-1) pathways, suggesting its therapeutic promise for RA [101]. Together, these studies offer new insights into necroptosis mechanisms and identify potential therapeutic strategies for RA by targeting key pathways involved in cartilage degeneration and inflammation (Figure 6) and (Table 2).

### 4.3. Significance of Necroptosis-Mediated Cartilage Degeneration in Trauma-Induced Arthritis

Cartilage injury triggers catabolic gene expression, inflammation, and cell death, leading to extracellular matrix degradation and increasing the risk of posttraumatic osteoarthritis (PTOA) [102,103,104]. Numerous studies have been conducted to identify critical mechanisms and therapeutic targets for managing necroptosis-mediated trauma-induced cartilage degeneration. Riegger and Brenner’s research highlighted necroptosis in human cartilage under trauma, emphasizing its link to inflammation and oxidative stress in OA. Their study found that Nec-1, an RIPK1 inhibitor, significantly protects against trauma-induced necroptosis, surpassing the caspase inhibitor zVAD. Nec-1 also reduces MLKL gene expression and prostaglandin E2 (PGE2) production, mitigating necroptotic cell death and inflammation. This research underscores the role of reactive oxygen species (ROS) in initiating RIPK1-mediated necroptosis and demonstrates that Nec-1 and NAC attenuate trauma-induced necroptosis, particularly with TNF/CHX stimulation, suggesting necroptosis becomes more prominent in later OA stages [105]. Tolberg-Stolberg et al. identified the necroptosis markers RIPK3 and MLKL in human OA cartilage, linked to PGE2 and nitric oxide (NO) release. They showed increased necroptosis in human and murine cartilage post-trauma, significantly inhibited by necrostatin-1. The release of DAMPs like HMGB1 and dsDNA from necroptotic cells suggests a role in inflammation and cell death, indicating that future research should focus on inhibiting necroptosis and DAMP release to prevent post-traumatic arthritis (PTA) [25]. Zhou et al. explored the later stages of necroptosis in TMJOA induced by mechanical force. Terminal deoxynucleotidyl transferase dUTP nick-end labeling (TUNEL) staining revealed significant chondrocyte death, with increased Caspase-8 indicating early apoptosis at four days, persisting but reduced at seven days. RIPK3 levels rose at seven days, indicating late-stage necroptosis. In the F + siRIPK1 group, both (Cas-8) Caspase-8 and RIPK3 levels decreased significantly, showing that RIPK1 inhibition reduces apoptosis and necroptosis in stress-induced TMJOA [88]. Zhang et al. found that mechanical stress induces necroptosis in chondrocytes, with RIPK1 peaking near the force source and TNF-α aligning with RIPK1 expression. Blocking Cas-8 did not increase necroptosis, suggesting complex mechanisms involving cytoplasmic calcium and ROS. Combined Nec-1 and Z-VAD treatment reduced TNF-α-induced ROS and necroptosis, highlighting RIPK1’s role in chondrocyte fate [106] Additionally, Coustry et al. demonstrated that D469del-(COMP) cartilage oligomeric matrix protein retention in chondrocytes triggers caspase-independent necroptosis through persistent endoplasmic reticulum (ER) stress, oxidative stress, and DNA damage, marked by increased NADPH oxidase 4 (Nox4), endoplasmic reticulum oxidoreductase 1 (Ero1), C/EBP Homologous Protein (Chop), growth arrest and DNA damage-inducible protein 34 (Gadd34), and the presence of cleaved apoptosis-inducing factor (tAIF) [107]. This collective research underscores the importance of targeting necroptosis pathways and associated markers to develop effective therapies for OA and TMJOA (Figure 7) and (Table 3).

## 5. Interactions with Other Regulated Cell Death Pathways

Cartilage degeneration is significantly influenced by chondrocyte death, particularly through necroptosis [105]. Exploring interrelated pathways is crucial for understanding this process. Sun, Kai, et al. found that inhibiting TRADD with ICCB-19 ameliorates chondrocyte necroptosis and OA by blocking the RIPK1-TAK1 pathway, crucial for necroptosis signaling. This inhibition reduces necroptosis and restores autophagy, promoting cellular health. In vitro, ICCB-19 reversed increased necroptosis and decreased autophagy markers in IL-1β-treated OA chondrocytes. In vivo, ICCB-19 reduced cartilage degeneration and necroptosis markers while restoring autophagy in a DMM-induced OA mouse model, highlighting autophagy’s protective role against necroptosis-induced damage [89]. Additionally, reducing RIPK1 expression in chondrocytes alleviates OA by disrupting the TRIF/MyD88-RIPK1-TRAF2 feedback loop. Lowering RIPK1 levels decreases necroptosis (RIPK3 and MLKL) and apoptosis markers (cleaved caspase 3), reducing cartilage damage and inflammation. This suggests that targeting RIPK1 could modulate both pathways for OA treatment [14]. Furthermore, RIPK1 activation can influence BMP7 expression levels, potentially creating a feedback loop for tissue repair or remodeling. However, elevated BMP7 can sensitize chondrocytes to necroptosis by enhancing necroptotic machinery, promoting cell death instead of repair. This dual role of BMP7 suggests it could amplify necroptotic signaling when combined with the activated necrosome (RIPK1-RIP3 complex), contributing to cartilage degeneration.

## 6. Targeting Necroptosis in Cartilage Degeneration: Promising Inhibitors and Therapeutic Potential

Necroptosis has been identified as a crucial factor in cartilage degeneration, and specific protein molecules mediate this process [108,109]. To address this, researchers have developed inhibitors targeting these regulatory proteins, aiming to identify new therapeutic agents. Notably, substances such as Nec-1, GSK’872, and NSA have shown significant inhibitory effects on necroptosis, thereby demonstrating potential in preventing cartilage degeneration [81,110,111,112,113]. Studies have indicated that these inhibitors can effectively reduce necroptotic cell death, highlighting their promise as treatment strategies for cartilage degeneration. The exploration of anti-necroptosis strategies in animal models supports the potential of these inhibitors to mitigate cartilage damage and inflammation by targeting necroptotic pathways. This approach underscores the importance of continuing to elucidate necroptosis inhibitors, as they could pave the way for innovative and targeted treatments for cartilage degeneration (Table 4).

### 6.1. RIPK1 Inhibitors

Nec-1, a selective allosteric inhibitor of RIPK1, binds to its kinase domain, stabilizing it in an inactive conformation [45,114,115,116,117]. Degterev et al. (2005) demonstrated Nec-1’s efficacy in inhibiting necrosis by binding it to the hydrophobic pocket of RIPK1’s kinase domain [118]. A study by Riegger and Brenner showed that Nec-1 significantly protects against trauma-induced necroptosis, surpassing the caspase inhibitor zVAD [105]. Additionally, GSK2982772, RIPA-56, VX-680, and MK-0457 are emerging RIPK1 inhibitors with the potential to treat necroptosis-related diseases [112,119,120]. AZD8330, identified by Qiu et al., activates cIAP1 to inhibit RIPK1-associated necrosis in OA, while KW2449, as demonstrated by Wang et al., ameliorates collagen-induced arthritis by inhibiting RIPK1-dependent necroptosis [92,100]. These findings highlight RIPK1 inhibition as a promising approach for developing therapies for cartilage degeneration. Furthermore, some potential inhibitors under investigation are those placed in the table (Table 4).

### 6.2. RIPK3 Inhibitors

GSK’872, a specific small molecule inhibitor of RIPK3 binding to its kinase domain, suppresses necroptosis and RIPK3-dependent inflammation [121,122,123]. GSK’840, GSK’843, and GSK’872 are RIPK3 kinase inhibitors, with GSK’843 and GSK’872 at higher concentrations potentially promoting RIPK1-dependent apoptosis and caspase 8 activation. GSK’840 offers superior specificity but cannot effectively inhibit murine RIPK3 [124,125]. Dabrafenib, a type I RIPK3 inhibitor, disrupts RIPK3-mediated MLKL phosphorylation and has clinical approval [45]. HS-1371, a type II RIPK3 inhibitor, targets RIPK3’s ATP-binding pocket. Clinical trial setbacks with RIPK3 inhibitors have questioned their efficacy in blocking necroptosis [119,126]. AZ-628, identified by Jeon J et al., effectively mitigates RIPK3-mediated OA pathogenesis, highlighting RIPK3 as a key target needing further research for preventing cartilage degeneration [24]. Furthermore, some potential inhibitors under investigation are those placed in the table (Table 4).

### 6.3. MLKL Inhibitors

NSA is the first compound reported to inhibit MLKL by targeting its N-terminal CC domain, acting as a specific downstream target for RIPK3 [56,115,127,128]. NSA prevents necroptosis in human cells by alkylating Cys86 in MLKL, a residue absent in murine MLKL, making it ineffective in murine models. Despite this, NSA shows protective effects in pulmonary I/R injury, spinal cord injury, and Alzheimer’s disease by suppressing necrosome formation [128,129]. Cartilage degeneration has a link to necroptosis, inflammation, oxidative stress, and cell death. Lee, Seung Hoon, et al. revealed that interferon-gamma (IFN-γ) can mitigate necroptosis and inflammation in RA by reducing MLKL production [90]. Riegger and Brenner noted that Nec-1, a RIPK1 inhibitor, protects against trauma-induced necroptosis and reduces MLKL gene expression and PGE2 production. Additionally, new MLKL inhibitors like GW806742X (SYN-1215) target the MLKL pseudokinase domain. Thioredoxin-1 (Trx1) can bind to MLKL, preventing its disulfide bond formation and polymerization, thus blocking necroptosis [78]. However, drugs that up-regulate Trx1 are still under investigation. While MLKL is a crucial target for inhibiting necroptosis, further research is needed to confirm these inhibitors’ efficacy in preventing cartilage degeneration. Furthermore, some potential inhibitors under investigation are those placed in the table (Table 4).

Additionally, novel inhibitors such as AZ-628, ICCB-19, the PLCγ1 inhibitor U73122, and green-lipped mussel (GLM) show potential as treatments for osteoarthritis (OA) by targeting necroptosis and inflammation. AZ-628 inhibits RIP3 and MLKL, stabilizes the chondrocyte microenvironment, reduces inflammation, and delays OA progression. ICCB-19 protects chondrocytes from TNF-α-induced damage by inhibiting TRADD signaling and necroptosis, reducing inflammation, ECM degradation, and apoptosis. U73122 reduces cartilage damage by inhibiting PLCγ1 and enhancing cartilage matrix synthesis. GLM exhibits anti-inflammatory and joint-protecting properties by modulating necroptosis markers RIP1, RIP3, and pMLKL in human OA chondrocytes. Furthermore, Zharp-99 demonstrates significant effectiveness in preventing necroptosis triggered by various necroptotic agents across human, mouse, and rat cell lines [89,90,130]. However, although the aforementioned compounds have shown significant potential in inhibiting necroptosis, their impact on other forms of cell death and cellular functions presents notable drawbacks that must be carefully considered before advancing to clinical applications (Table 4).

## 7. Conclusion and Future Direction of Preventing Necroptosis

Cartilage degeneration, a global health concern causing significant disability, has been increasingly linked to necroptosis, a form of programmed cell death distinct from apoptosis [25,34,78]. This review has explored the significance of necroptosis in cartilage degeneration, detailing its occurrence, mechanisms, and implications for therapeutic intervention. Key markers such as RIPK1, RIPK3, and MLKL, along with critical pathways like the TNFα-Syndecan 4-TNFα vicious cycle, BMP7, and the TRIM24-RIP3 axis, have been identified in necroptosis-mediated cartilage degeneration. External compressive forces, trauma, OA, and RA are the primary factors influencing this process. Recent findings suggest that inhibition of TRADD can reduce chondrocyte necroptosis and alleviate OA symptoms, while the combined inhibition of PLCc1, apoptosis, and necroptosis promotes cell survival. Notably, the TRIF/MyD88-RIPK1-TRAF2 negative feedback loop, upregulation of cIAP1, and regulation of GLM, ASICs, and interferon-gamma are also involved. Furthermore, HMGB1/MCP1/Chitotriosidase I-mediated necroptosis, ROS, and mutations such as D469del-COMP contribute to this complex mechanism. Understanding these pathways provides insight into necroptosis-driven cartilage degeneration and highlights potential therapeutic targets.

Despite progress in therapies targeting necroptosis for OA, RA, and trauma-induced necroptosis, gaps remain in understanding their mechanisms, long-term efficacy, safety, and potential off-target effects. Agents such as Nec-1, amiloride, IFN-γ, KW2449, and irisin show promise but require further investigation. The interplay between apoptosis and necroptosis, including the roles of MLKL, RIPK1, and caspase 8, remains unclear. RIPK1 activation affects BMP7 expression, establishing a feedback loop that can enhance tissue repair. However, elevated BMP7 levels may sensitize chondrocytes to necroptosis, leading to increased cell death and cartilage degeneration through the RIPK1-RIPK3 complex. While BMP7 is essential for cartilage remodeling, its inhibition might negatively impact this process. Further research is needed to balance BMP7’s roles in necroptosis and cartilage repair. In chondrocytes with the D469del-COMP mutation, ER stress initially triggers autophagy to manage misfolded COMP proteins. Persistent ER stress eventually shifts to necroptosis, increasing cell death and contributing to cartilage degeneration in OA and TMJOA. The interplay between autophagy and necroptosis in this context remains underexplored. High doses of the PLCγ1 inhibitor U73122 may increase cell death through apoptosis and necroptosis, undermining its benefits on matrix synthesis. Therefore, careful dosing and combination with inhibitors like Z-VAD and Nec-1 are essential to maximize therapeutic benefits while minimizing adverse effects. Future clinical trials are needed to better validate necroptosis as a therapeutic target and improve treatment strategies.

## Figures and Tables

**Figure 1 biomolecules-14-01192-f001:**
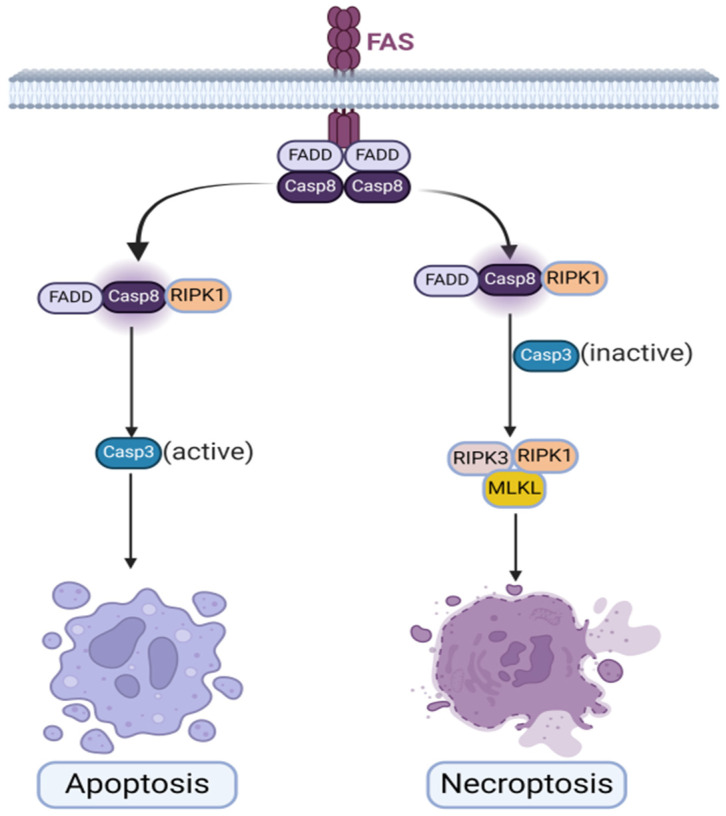
The graphical presentation of common initiation executes significantly different pathways of apoptosis and necroptosis.

**Figure 2 biomolecules-14-01192-f002:**
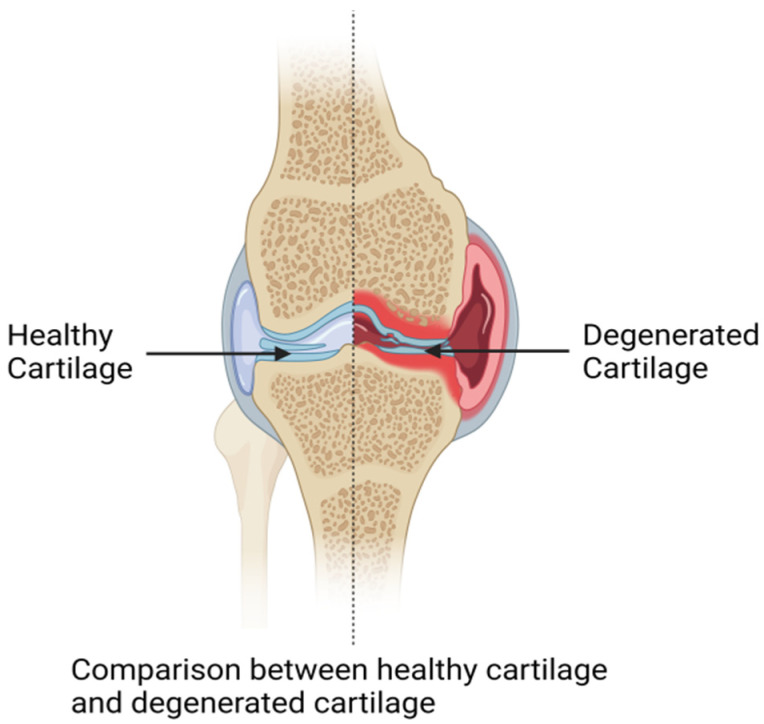
A comparison between healthy and degenerated cartilage in the knee joints.

**Figure 3 biomolecules-14-01192-f003:**
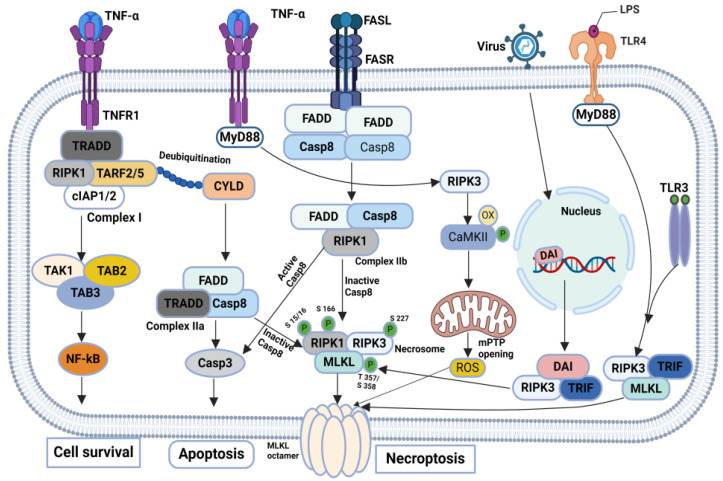
Typical molecular mechanisms of necroptosis: Necroptosis-mediated cell death is initiated by the activation of death receptors such as the TNF receptor, FAS, Toll-like receptor, or interferon receptor. These receptors activate RIPK1 or other RIP homologous interaction motif (RHIM) domain-containing proteins, which then interact with RIPK3 to form the necrosome complex. RIPK3 is activated through phosphorylation and subsequently phosphorylates MLKL. Phosphorylated MLKL oligomerizes and moves to the plasma membrane, triggering necroptosis. RIPK1, RIPK3, and MLKL are the core components of TNF-induced necroptosis. Additionally, RIPK3 can activate CaMK II, leading to the opening of the mPTP and necroptosis in cardiomyocytes. RIPK3 can also be activated by other RHIM domain-containing proteins like TRIF and DAI, expanding the mechanisms of RIPK3.

**Figure 4 biomolecules-14-01192-f004:**
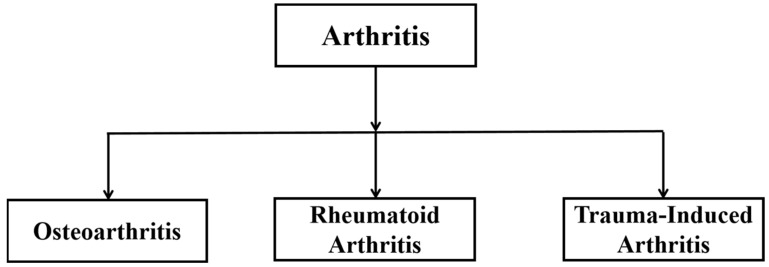
Categorization of arthritis.

**Figure 5 biomolecules-14-01192-f005:**
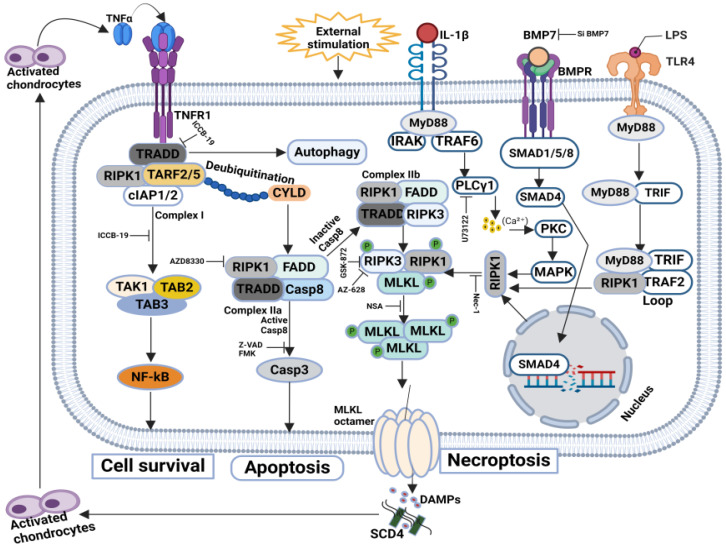
**Molecular mechanisms of necroptosis in OA and TMJOA**: In TMJOA, TNFα induces Syndecan 4 (SDC4), which amplifies TNFα signaling and triggers necroptosis, releasing cartilage-degrading enzymes and intensifying inflammation. Inhibiting RIPK3, pMLKL, and SDC4 protects cartilage and reduces inflammation. BMP7 induces necroptosis through RIP1, with BMP7 silencing reducing RIPK1-induced necroptosis and restoring ECM gene expression. High RIPK3 expression accelerates cartilage degradation, while RIPK3 inhibition by AZ-628 mitigates OA progression. TRADD inhibition with ICCB-19 blocks the RIPK1-TAK1 pathway, reducing inflammation and necroptosis. PLCγ1 inhibition, combined with apoptosis and necroptosis blockers, enhances cartilage matrix synthesis. RIPK1 knockdown disrupts the TRIF/MyD88-RIPK1-TRAF2 pathway, alleviating OA. AZD8330 activates cIAP1, inhibiting RIPK1-associated necrosis, and preserving cartilage.

**Figure 6 biomolecules-14-01192-f006:**
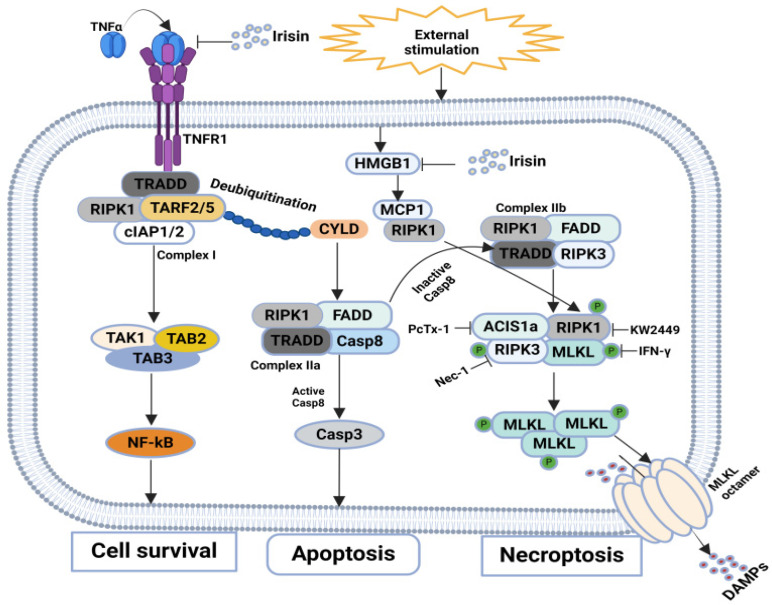
**Molecular mechanisms of necroptosis in RA:** Nec-1 and amiloride inhibit necroptosis in RA chondrocytes by targeting the RIP1/RIP3/p-MLKL pathway, with ASIC1a-mediated upregulation reversible by PcTx-1 or Nec-1. IFN-γ mitigates necroptosis and inflammation by reducing MLKL and modulating inflammatory responses, despite its proinflammatory role. KW2449 ameliorates collagen-induced arthritis by inhibiting RIPK1-dependent necroptosis, reducing RIPK1 and MLKL levels. Irisin reduces necroptotic signaling and inflammation via the NF-kB and Nrf2/HO-1 pathways, downregulating TNF-α, MCP1, and HMGB1, promoting chondrocyte recovery.

**Figure 7 biomolecules-14-01192-f007:**
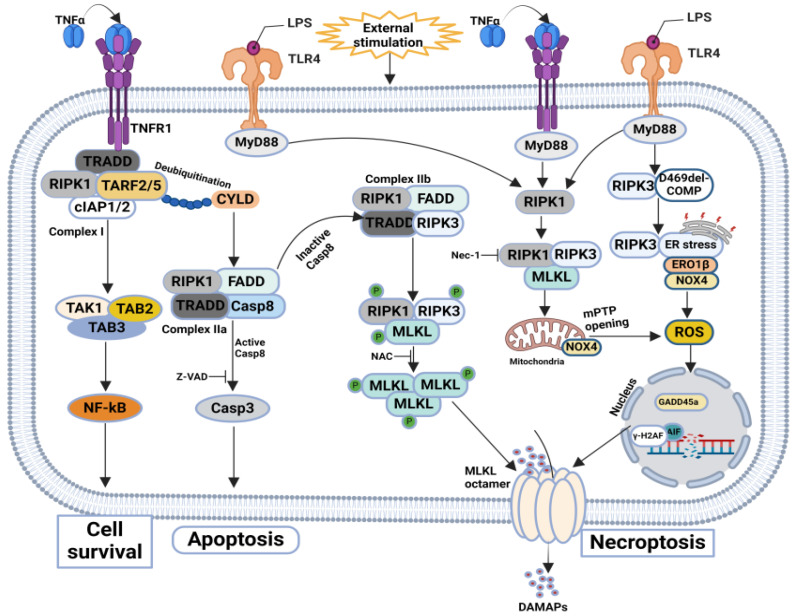
**Molecular mechanisms of necroptosis in trauma:** Nec-1, a RIPK1 inhibitor, surpasses zVAD in protecting against trauma-induced necroptosis by reducing MLKL expression and PGE2 production. ROS are crucial in RIPK1-mediated necroptosis, which is more prominent in late-stage OA. Necroptosis markers RIPK3 and MLKL in OA cartilage are linked to PGE2 and NO release, and necrostatin-1 inhibits post-trauma necroptosis. In TMJOA, RIP1 inhibition reduces apoptosis and necroptosis. Mechanical stress induces necroptosis in chondrocytes, with Nec-1 and Z-VAD reducing TNF-α-induced ROS and necroptosis. D469del-COMP retention triggers necroptosis via ER stress, oxidative stress, and DNA damage.

**Table 1 biomolecules-14-01192-t001:** Key findings and experimental model to identify necroptosis-mediated cartilage degeneration in OA.

Study Focus	Key Findings	Experimental Model	Reference
TRIF/MyD88-RIPK1-TRAF2 feedback loop and its involvement in necroptosis	Decreased RIPK1 expression modulates the TRIF/MyD88-RIPK1-TRAF2 feedback loop to reduce inflammation.	OA models: RIPK1 knockdown reduced cartilage damage and inflammation.	[14]
BMP7 and its involvement in necroptosis	BMP7, induced by RIPK1, drives chondrocyte necroptosis and cartilage damage.	OA models: overexpression of BMP7 led to increased necroptosis and cartilage loss.	[86]
TRIM24-RIP3 and its involvement in necroptosis	TRIM24-RIP3 axis regulates OA progression; AZ-628 reduces OA severity.	Mouse model: AZ-628 treatment significantly reduced OA severity and cartilage damage.	[87]
SDC4 and its involvement in necroptosis	TNF-α activates SDC4, which amplifies inflammation via RIPK3 and pMLKL, leading to necroptosis and cartilage degradation.	Unilateral arthritis condition (UAC) model: targeting RIPK3, pMLKL, and SDC4 reduced necroptosis and protected cartilage.	[88]
RIPK1-TAK1 pathway and its involvement in necroptosis	ICCB-19 inhibits TRADD, blocks the RIPK1-TAK1 pathway, and restores autophagy.	DMM mouse model: ICCB-19 reduced necroptosis, inflammation, and cartilage degeneration.	[89]
PLCγ1and its involvement in necroptosis	U73122 inhibits PLCγ1 and, combined with apoptosis and necroptosis inhibitors, enhances cartilage matrix synthesis.	IL-1β-treated rat chondrocytes: U73122 improved cartilage matrix synthesis and reduced damage.	[90]
RIPK1 and its involvement in necroptosis	RIPK1 inhibition protects TMJOA cartilage by suppressing inflammatory signaling and necroptosis.	TMJOA models: RIPK1 inhibition reduced cartilage degradation and inflammatory factors.	[91]
cIAP1 and its involvement in necroptosis	AZD8330 activates cIAP1 and inhibits RIP1-associated necrosis pathway, preventing chondrocyte necroptosis.	DMM mouse model: AZD8330 significantly reduced necroptosis markers and preserved cartilage integrity.	[92]

**Table 2 biomolecules-14-01192-t002:** The key findings and experimental model used to identify necroptosis-mediated cartilage degeneration in RA.

Study Focus	Key Findings	Experimental Model	Reference
ASICs and their involvement in necroptosis	ASIC1a mediates necroptosis through the RIPK1/RIPK3/p-MLKL pathway in RA cartilage injury.	Adjuvant arthritis (AA) rat model: Nec-1 and amiloride inhibit acid-induced chondrocyte necroptosis in RA, reducing necroptosis markers (RIPK1, RIPK3, p-MLKL, PGAM5) and inflammatory cytokines (TNF-α, IL-1β) to protect cartilage. ASIC1a binds RIPK1 upon acidosis, confirmed by immunofluorescence. Both inhibitors alleviate cartilage injury, with RIPK1 and RIPK3 upregulation via ASIC1a reversible by PcTx-1 or Nec-1.	[98]
IFN-γ and their involvement in necroptosis	IFN-γ attenuates necroptosis by decreasing MLKL production. IFN-γ reduces inflammation by inhibiting necroptosis, despite its known proinflammatory role.	IFN-γ reduces cartilage damage in CIA mice and RA patients by downregulating cFLIPL and inhibiting necroptosis mediators (RIPK1, RIPK3, MLKL). Its absence worsens inflammation via STAT3 activation and increases TNF-α/IL-17 levels.	[99]
KW2449 and their involvement in necroptosis	KW2449 ameliorates CIA by inhibiting RIPK1-dependent necroptosis.	KW2449 effectively reduces joint swelling, arthritis scores, and plasma cytokines compared to MTX in CIA rats. It inhibits RIPK1-mediated necroptosis, reducing RIPK1 and MLKL levels, and enhances cell viability.	[100]
Irisin and their involvement in necroptosis	Irisin reduces necroptotic signaling and inflammation in RA.	Downregulated TNF-α, MCP1, and HMGB1; anti-inflammatory and antioxidant effects through NF-kB and Nrf2/HO-1 pathways.	[101]

**Table 3 biomolecules-14-01192-t003:** Key findings and experimental model to identify necroptosis-mediated cartilage degeneration in trauma.

Study Focus	Key Findings	Experimental Model	Reference
Role of necroptosis in PTA and OA	Increased necroptosis post-trauma in OA cartilage. Necroptosis causes inflammation via cellular content release. RIPK1 inhibitor necrostatin-1 reduces necroptosis. Release of DAMPs perpetuates inflammation.	In human and murine cartilage models, RIPK3 and MLKL markers linked to PGE2 and NO release, consistent necroptosis marker expression in murine models, and variable marker expression in human samples due to trauma and fixation timing.	[25]
Necroptosis in late-stage TMJOA and effects of RIP1 inhibition	Significant chondrocyte death post-mechanical force. Increased Caspase-8 at 4 days, reduced at 7 days. Increased RIP3 at 7 days.	Mechanical force application on TMJOA, TUNEL staining for chondrocyte death, Caspase-8 and RIP3 expression analysis, decreased apoptosis and necroptosis in F + siRIP1 group.	[88]
Role of necroptosis in OA, and oxidative stress	Nec-1 protects against trauma-induced necroptosis, ROS initiates RIPK1-mediated necroptosis, Nec-1 and NAC attenuate necroptosis	Human cartilage subjected to trauma, reduced MLKL gene expression and PGE2 production, attenuation of TNF/CHX-induced p-MLKL-positive cells, potential increase in necroptosis in later OA stages.	[105]
Role of necroptosis in TMJOA and mechanisms involving calcium and ROS	Mechanical stress induces chondrocyte necroptosis, RIP1 peaks and normalizes within 7 days, Caspase-8 blockade did not increase necroptosis, Nec-1 and Z-VAD reduce TNF-α-induced necroptosis.	Mechanical stress on chondrocytes, TNF-α peaks 4 days post-force, combined Nec-1 and Z-VAD treatment mitigated ROS and necroptosis, normalization of RIP1, RIP3, and Caspase-8 levels by 7 days.	[106]
Effects of D469del-COM induces necroptosis in chondrocytes	D469del-COMP retention induces necroptosis, increased Chop and Gadd34 expression, high Nox4 and Ero1 levels increased the ROS and DNA damage, as indicated by Gadd genes and H2AX.	After 4 days mRNA expression, 5 days without inducing agent, there was stimulation of ER stress markers, increased ROS and oxidative stress, presence of tAIF, and absence of activated caspases	[107]

**Table 4 biomolecules-14-01192-t004:** Promising inhibitors and therapeutic potential.

Name of Inhibitors	Mechanism of Action	Potential Benefits	Side Effects	Clinical Trial
Necrostatins (Nec-1, Nec-7, Nec-1s)	RIPK1 inhibitors prevent necroptosis by blocking RIPK1-RIPK3 complex formation	Effective in inhibiting necroptosis; Nec-1s lacks indoleamine 2,3-dioxygenase (IDO) pathway inhibition	Nec-1 inhibits IDO; potential off-target effects	Clinical trials ongoing, particularly in cancer therapy
GSK’963, GSK’872	GSK’963 targets RIPK1, GSK’872 inhibits RIPK3	Fewer off-target effects, promising in inflammatory disease models	Long-term outcomes unclear; GSK’872 induces apoptosis at high doses	Ongoing clinical trials for necroptosis-related diseases
RIPA-56	Selective RIPK1 inhibitor, similar to necrostatins	Prevents disease progression in multiple sclerosis models	Potential cytotoxicity concerns	Clinical trials ongoing for necroptosis-related conditions
Furo[2,3-d]pyrimidines	Selective RIPK1 inhibitors	Potent anti-inflammatory effects; potential in cartilage degeneration	More research needed for cartilage-related pathologies	Under investigation in necroptosis-related conditions
VX-680, MK-0457	Inhibitors of Aurora kinase, involved in cell division and necroptosis	Effective in cancer models, leukemia, and ovarian cancer	Strong cardiac toxicity, cytotoxic effects	Clinical trials suspended (MK-0457); further research needed
Dabrafenib, HS-1371, AZ-628	RIPK1/RIPK3 inhibitors	Reduce necroptosis and protect against liver injury	Off-target kinase effects, including skin reactions and cancer risks	Clinical trials ongoing for ischemic injury and related conditions
U73122	Modulate cell death pathways, U73122 is a PLC inhibitor	Potential in modulating cell death and calcium signaling	Complex effects on cell death pathways	Further exploration needed
DCC2036	DCC-2036 directly inhibits RIPK1 and RIPK3 kinase activities (R53)	Potential in reducing chondrocyte damage and inflammation	Off-target effects in rapidly dividing cells	Clinical trials ongoing for cartilage degeneration and arthritis
PcTx-1, amiloride	Inhibitors of ASIC and sodium channels	Reduce inflammation and necroptosis	Amiloride may cause electrolyte imbalances	Requires further research
ICCB-19	Target oxidative stress and TRADD	Reduce oxidative stress and necroptosis	Prolonged ROS inhibition may impair normal signaling	Not yet advanced to human clinical trials
TUDCA	Anti-necroptotic bile acid derivative	Well-tolerated, effective in protecting cells from various stressors	Minimal side effects	Ongoing trials, need specific trials for necroptosis-related conditions
Hydroxyanisole, DPI	Antioxidants inhibit NADPH oxidase and reduce ROS	May reduce oxidative stress and apoptosis	Potential disruption of mitochondrial function	Undergoing phase II/III trials for oxidative stress-related conditions
GLM	RIPK1, RIPK3, and MLKL inhibitors	Reduced the expression of RIPK1, RIPK3, MLKL, and anti-inflammatory response	Not known	Requires further research
Thioredoxin-1 (Trx1)	MLKL inhibitors	MLKL disulfide bond formation and polymerization	Not known	Requires further research
KW2449	Phosphorylated RIPK1	Potential in reducing chondrocyte damage and inflammation	Not known	Requires further research
Zharp-99	Inhibitor of RIPK3 kinase activity	Significantly ameliorates TNF-induced systemic inflammatory response syndrome (SIRS) in mouse model.	Not known	The starting point for development, requires further investigation

## Data Availability

The authors confirm that the data supporting the findings of this study are available within the article.

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
