# Peer review of "Significance of Necroptosis in Cartilage Degeneration"

_biomolecules, 2024, doi:10.3390/biom14091192_

Round 1
Reviewer 1 Report
Comments and Suggestions for Authors
- The manuscript is comprehensive in terms of looking at the role of necroptosis in cartilage degeneration and its implications in OA and RA. The authors highlighted the significance of necroptosis and its potential as a therapeutic target. However, the manuscript could benefit from addressing some gaps and with a bit more depth in the discussion in some areas.
Suggestions:
- While the review is thorough, you may want to consider including more critical analysis and synthesis of the data. If possible, integrate novel insights or hypotheses that could drive future research in the field. This suggestion is with an eye towards adding a bit more originality and impact to the manuscript.
- The manuscript mainly focuses on OA and RA, but necroptosis is relevant in other conditions, such as trauma-induced arthritis and autoimmune diseases. You may want to expand the discussion to include these conditions to make the review a bit more comprehensive and applicable to a broader audience.
- The manuscript heavily emphasizes necroptosis while giving less attention to other forms of cell death, such as apoptosis and autophagy, which also play significant roles in cartilage degeneration. A more balanced discussion of these pathways would provide a holistic view of the mechanisms involved in cartilage degradation.
- The discussion of the molecular mechanisms underlying necroptosis could be expanded. You may want to include more detailed descriptions of how key proteins like RIPK1, RIPK3, and MLKL interact and contribute to the necroptotic process to enhance the manuscript's scientific rigor.
- You may want to check some studies cited in the manuscript as they seem to present conflicting evidence regarding the role of necroptosis in cartilage degeneration. It would be beneficial to address these discrepancies directly and offer potential explanations or suggestions for future research to resolve these conflicts.
- While the manuscript touches on potential therapeutic targets, it would benefit from a more in-depth exploration of the challenges associated with translating these findings into clinical practice. You may consider discussing the potential side effects, efficacy, and feasibility of these therapies in a bit more detail.
- The manuscript contains some repetitive sections where similar information is presented multiple times. You may want to streamline these sections to improve the manuscript's readability and overall flow.
- It maybe helpful to include more visual aids, such as diagrams that illustrate the molecular pathways involved in necroptosis and its comparison with other cell death mechanisms, would help clarify complex concepts and enhance the reader's understanding.
- Consider including more recent studies to ensure the review is up-to-date with the latest advancements in the field. This will also demonstrate that the authors are aware of and have considered the most current research in their analysis.
- Some sections of the manuscript could be made more concise. Consider revising the text to eliminate unnecessary jargon and enhance clarity, making it more accessible to a broader scientific audience.
The quality of English in the manuscript is generally strong, with clear and precise use of terminology that is appropriate for the subject matter. However, there are areas where the language could be refined to enhance readability and clarity. Some sections contain redundant information that could be streamlined, and a few sentences are overly complex, which might benefit from being broken down into shorter, more concise statements. Additionally, ensuring consistent use of terminology throughout the manuscript would improve coherence. Overall, while the English is correct, these adjustments would make the manuscript more accessible and improve the overall flow of the text.
Author Response
We highly appreciate your esteemed suggestions and comments, as well as the time you took to provide feedback. I have attached the reviewer's response for your reference. Please find the attached file.

Reviewer 2 Report
Comments and Suggestions for Authors
The topic is current and of great biomedical interest. It presents some well-designed and easy-to-understand illustrations. I believe that this is an interesting review for readers of Biomolecules. I have no significant objections to the manuscript.
Author Response
We greatly appreciate your esteemed comments and the time you took to provide feedback.
Reviewer 3 Report
Comments and Suggestions for Authors
Authors provided a comprehensive review of role of Necroptosis in Arthritis (OA/RA) and provided a detailed mechanism of action (MoA) well supported with figures, tables and references. Overall, review is well organized, easy to read and follow on authors ideas.
If authors have copied some figures especially the cell signaling for explaining MoA, authors should cite the resource/reference for that figure.
Comments on the Quality of English Language
Minor typos and grammatical errors.
Author Response

(The authors gave the same response as above.)
